# Detection of Citrus Huanglongbing Based on Multi-Input Neural Network Model of UAV Hyperspectral Remote Sensing

**Xiaoling Deng** [1,2,†], **Zihao Zhu** [2,3,†], **Jiacheng Yang** [1,2], **Zheng Zheng** [4], **Zixiao Huang** [3], **Xianbo Yin** [1,2], **Shujin Wei** [1] **and Yubin Lan** [1,2,*]

1   College of Electronic Engineering, South China Agricultural University, Guangzhou 510642, China; dengxl@scau.edu.cn (X.D.); yangjc@stu.scau.edu.cn (J.Y.); yxb@stu.scau.edu.cn (X.Y.); shujinwei@stu.scau.edu.cn (S.W.)
2   National Center for International Collaboration Research on Precision Agricultural Aviation Pesticide Spraying Technology, Guangzhou 510642, China; zhuzihao@stu.sau.edu.cn
3   College of Engineering, South China Agricultural University, Guangzhou 510642, China; zixiaohuang@stu.scau.edu.cn
4   College of Agriculture, South China Agricultural University, Guangzhou 510642, China; zzheng@scau.edu.cn
*   Correspondence: ylan@scau.edu.cn; Tel.: +86-139-2270-7507
†   Both authors contributed equally to this work.

**Abstract:** Citrus is an important cash crop in the world, and huanglongbing (HLB) is a destructive disease in the citrus industry. To efficiently detect the degree of HLB stress on large-scale orchard citrus trees, an UAV (Uncrewed Aerial Vehicle) hyperspectral remote sensing tool is used for HLB rapid detection. A Cubert S185 (Airborne Hyperspectral camera) was mounted on the UAV of DJI Matrice 600 Pro to capture the hyperspectral remote sensing images; and a ASD Handheld2 (spectrometer) was used to verify the effectiveness of the remote sensing data. Correlation-proven UAV hyperspectral remote sensing data were used, and canopy spectral samples based on single pixels were extracted for processing and analysis. The feature bands extracted by the genetic algorithm (GA) of the improved selection operator were 468 nm, 504 nm, 512 nm, 516 nm, 528 nm, 536 nm, 632 nm, 680 nm, 688 nm, and 852 nm for the HLB detection. The proposed HLB detection methods (based on the multi-feature fusion of vegetation index) and canopy spectral feature parameters constructed (based on the feature band in stacked autoencoder (SAE) neural network) have a classification accuracy of 99.33% and a loss of 0.0783 for the training set, and a classification accuracy of 99.72% and a loss of 0.0585 for the validation set. This performance is higher than that based on the full-band AutoEncoder neural network. The field-testing results show that the model could effectively detect the HLB plants and output the distribution of the disease in the canopy, thus judging the plant disease level in a large area efficiently.

**Keywords:** hyperspectral remote sensing; huanglongbing; UAV; multi-input

## 1. Introduction

Citrus is one of the most cultivated fruits in the world, and it is also one of the most widely grown and most economically important fruit crops in southern China. Citrus is very susceptible to pests and diseases, causing considerable economic losses [1]. Citrus huanglongbing (HLB), which is a bacterial disease caused by *Candidatus Liberibacter asiaticus,* is generally considered to be the most complex citrus disease with a severe impact on the citrus industry because of its rapid spreading power, huge destructive power, and incurability. At present, the effective measures to control HLB

are to uproot the entire affected plant. Therefore, it is particularly important to detect HLB as soon as possible [2].

The symptoms of HLB are varied, ranging from uniform yellowing, mottled yellowing, lack of element yellowing, and overall yellowing to withering. Among them, mottled yellowing is the most typical symptom of HLB. The fruits of the infected plants show the 'green fruit disease' or the end of the fruit is red because of the abnormal colour change of the fruit. In addition to the symptoms visible to the naked eye, HLB causes microscopic changes in the plant physiology. These changes can be observed with the aid of external equipment and make it feasible to use map technology to detect HLB [3,4].

At present, the diagnostic methods of HLB are mainly field analysis and laboratory biochemical analysis. Field analysis is the fastest method for diagnosing HLB. It is a simple and easy method to use and does not require equipment assistance [5]. However, this method is subjective, requires relatively high knowledge and experience and has considerably varying accuracy. Laboratory biochemical analyses, including pathogenic microscope observation, biochemical index detection, nucleic acid probe detection, and Polymerase Chain Reaction (PCR) detection, are relatively complicated, with a long detection period and require considerable professional knowledge. For large-scale citrus orchards, these methods incur significantly high economic and time costs, which is not conducive to the promotion of actual agricultural production [6].

With the rapid development of computer technology and machine learning technology, information technology has made considerable progress in the detection of HLB. Li et al. [7] explored the possible use of satellite remote sensing to detect HLB through WorldView-2 satellite imagery. Pourreza et al. [8] and Deng et al. [9,10] adopted visible light images combined with the machine learning technology to detect HLB and achieved good accuracy under the set conditions, but could not solve the problem of metamerism. Wetterich et al. [11] and Deng et al. [12] used fluorescent devices combined with machine learning methods to detect HLB and achieved high accuracy.

With the development of artificial intelligence technology and intelligent manufacturing technology, drones have been widely used in many industries [13]. The UAV remote sensing technology has been gradually applied to the detection of diseases and insect pests in agricultural areas [14,15]. With the rich spectrum and image information of large-area farmland, hyperspectral remote sensing technology is expected to vigorously promote the development of smart agriculture. Yang et al. [16] combined multispectral and hyperspectral remote sensing to detect cotton root rot and verified the feasibility of using hyperspectral remote sensing to detect early root rot diseases. Wong et al. [17] used hyperspectral images to detect the effect of HLB on the citrus fruit quality in different seasons. Yue et al. [18] used UAV hyperspectral remote sensing to measure and estimate the plant height and above-ground biomass (AGB) of wheat. Abdulridha et al. [19] used drone-based hyperspectral remote sensing and the indoor hyperspectral detection of citrus canker and compared the detection accuracy of 31 vegetation indices to citrus canker. It has been found that the high-precision recognition of citrus canker can be achieved indoors or in air. Garza et al. [20] used RGB UAV remote sensing to detect citrus HLB and foot rot and explained the subtle differences in tree health caused by various diseases through the triangular green index (TGI).

At present, ground hyperspectral technology has made certain progress in the detection of HLB. Mei et al. [21] collected the hyperspectral images of citrus leaves with different degrees of health, zinc deficiency, and three infections of HLB in the laboratory environment, and adopted the PLS–DA model to construct an early discrimination model of HLB with a classification accuracy rate of approximately 96.4%. Liu et al. [22,23] analysed the hyperspectral data of citrus leaves with respect to health, vitamin deficiency, and yellow dragon disease in a laboratory environment, by using a series of methods to extract the feature variables and build the classification model with high accuracy. Deng et al. [24] carried out the HLB detection research by using both an indoor hyperspectral imager and an outdoor spectrometer. Different symptoms, including healthy, symptomatic, and asymptomatic categories, can be discriminated by using different machine learners. Li et al. [25] collected citrus leaf

hyperspectral data in the laboratory and field environments and obtained the red edge position (REP) of the citrus leaf spectrum by the first-order differential, exploring the characteristics of HLB REP and the degree to which REP can distinguish HLB. Sankaran et al. [26,27] collected multispectral cameras and thermal imaging cameras on the ground mobile platform to collect citrus near-ground canopy image data, revealing that the feature bands were 560 nm and 710 nm, with 87% accuracy by using SVM. Mishra et al. [28,29] discovered the feature bands of citrus leaves at the top of citrus disease trees and constructed a vegetation index, combined with SVM or the weighted nearest neighbour algorithm, can achieve more than 90% classification accuracy. The above studies have shown the feasibility of using hyperspectral images from the ground for HLB detection. However, the heavy workload of ground-based hyperspectral detection is not conducive to large-scale orchards.

In Florida, USA, Kumar et al. [30], Francisco et al. [31], Li X et al. [32], and Li H et al. [33] used a UAV or aircraft to perform hyperspectral remote sensing on citrus orchards, using machine learning and other algorithms to construct HLB discriminant models, with relatively low accuracy. In Guangdong Province, China, Lan et al. [34,35] carried out UAV remote sensing detection on citrus orchards, using hyperspectral remote sensing and multispectral remote sensing techniques to realise the remote sensing detection of HLB with relatively high accuracy. However, the models constructed by the above research do not combine the physiological characteristics of vegetation and cannot identify citrus plants with different disease levels. Therefore, the purpose of this study was to realise HLB disease detection, combined with the spectral characteristics of vegetation, by using the UAV hyperspectral remote sensing technology in a large-scale area.

## 2. Materials and Data Pre-Processing Methods

### 2.1. Data Collection

The citrus orchard test area selected for this study is located in Huizhou City, Guangdong Province, China (N23°29′57.81″–N23°29′59.31″, E114°28′8.39″–E114°28′12.26″, 40 m). The climate in this area is mild and humid, suitable for planting citrus and other fruit trees.

The citrus varieties in the test area are all *Citrus Reticulate Blancocv Shangtanju*, which are in the mature stage. In all, 334 citrus fruit trees were planted in nine rows in the test area. The citrus fruit trees were divided into two categories: healthy (H) and HLB (D). The citrus fruit trees infected with HLB were further divided into D1 and D2 according to the HLB symptom level in the canopy, where D1 implied that less than 50% of the treetops of the citrus tree showed HLB symptoms and D2 indicated that more than 50% of the treetops showed HLB symptoms. According to the field survey results, there were 293 H citrus trees and 41 HLB citrus trees, including 22 D1 grade citrus trees and 19 D2 grade citrus trees. Figure 1 shows the location and cultivation of the study site.

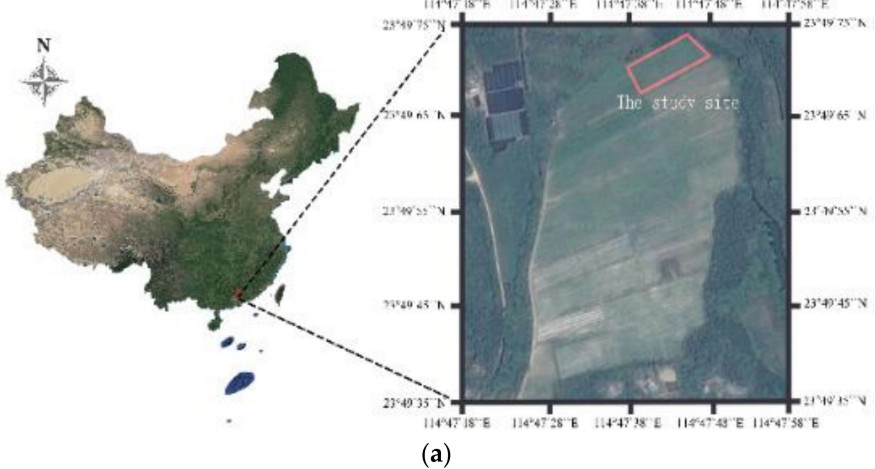

(**a**)

**Figure 1.** *Cont.*

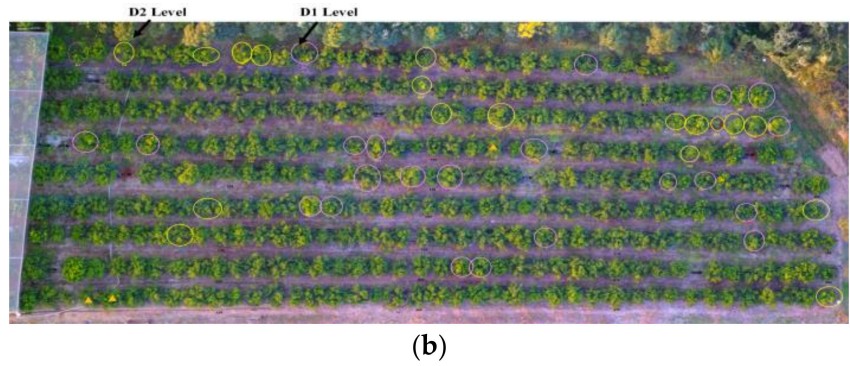

(**b**)

**Figure 1.** (**a**) Location of the study site; (**b**) investigation of the citrus orchard.

In this study, an ASD spectrometer was used to collect the ground hyperspectral data, and Cubert S185 mounted on a UAV of DJI Matrice 600 Pro was used to capture low-altitude hyperspectral remote sensing images at a flight height of 60 m and a flight speed of 4–5 m/s. The hyperspectral remote sensing data from Cubert S185 were mainly used to analyse and build the HLB detection models, and the hyperspectral data from ASD were mainly used to evaluate the data quality of the UAV remote sensing images. Figure 2a shows the UAV remote sensing acquisition system, including a flight platform, an electric gimbal, a hyperspectral data acquisition system, and a ground station control system. A diffuse reflection grayboard measuring 60 cm × 60 cm in size was placed on the flat ground of the test area, which was imaged in the UAV hyperspectral imagery, and radiation correction on the hyperspectral camera was performed in advance.

Figure 2b shows the ground data collecting method, where the ground hyperspectral acquisition device used was a convenient handheld non-imaging ground object spectrometer ASD FieldSpec HandHeld 2 (hereinafter referred to as HH2). Under the guidance of HLB experts and according to the PCR detection results, all of the HLB plants in the field were labelled as H plants and HLB plants correctly. When the UAV flew over the grayboard, HH2 collected the hyperspectral data of the grayboard at the same time. Table 1 shows the parameter information of Cubert S185 and ASD HH2.

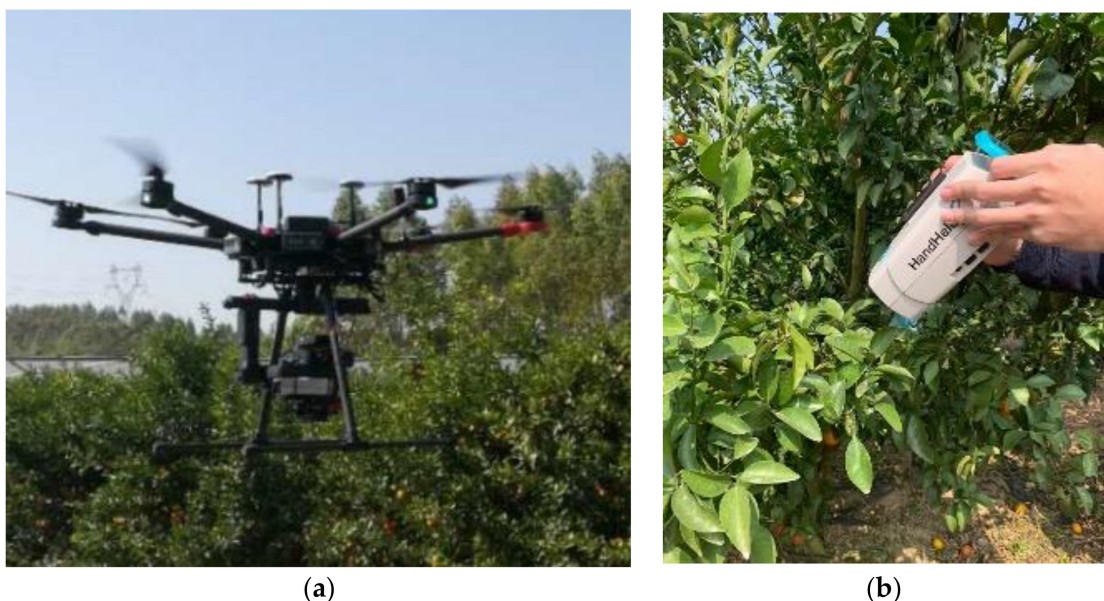

(**a**)                    (**b**)

**Figure 2.** Hyperspectral remote sensing device: (**a**) UAV (Uncrewed Aerial Vehicle) remote sensing device; (**b**) ground hyperspectral data collection device.

**Table 1.** Parameter information of Cubert S185 and HH2.

| Name | Spectral Range (Nm) | Sampling Interval (Nm) | Spectral Resolution (Nm) | Channel Number | Data Type | Size (Mm) | Weight (Kg) |
|---|---|---|---|---|---|---|---|
| Cubert S185 | 450–950 | 4 | 8@532 | 125 | Grayscale image pixel 1000 × 1000 Hyperspectral pixels 50 × 50 | 195 × 67 × 60 | 0.47 |
| HH2 | 325–1075 | 1 | <3.0 @700 | 700 | Spectral reflectance | 90 × 140 × 215 | 1.2 |

### 2.2. UAV Hyperspectral Data Quality Evaluation

Table 1 shows that the wavelength range of HH2 and S185 were different. There were 750 bands in HH2 and 125 bands in S185. For a better comparison to evaluate the UAV hyperspectral data quality from HH2, the data of HH2 were resampled, as shown in Figure 3.

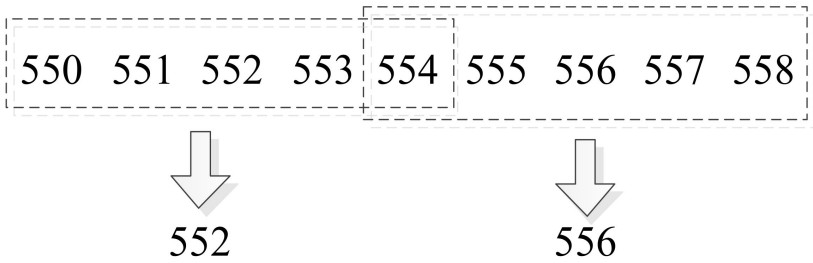

**Figure 3.** Illustration of resampling.

The hyperspectral data of the grayboard, which was placed in the test area, captured by S185 and HH2, were compared and analysed. Furthermore, the average spectra of the canopy of the H citrus plants and the HLB plants were compared between S185 and HH2. Figure 4a shows the grayboard spectrum, and Figure 4b shows the spectra of the canopy acquired by S185 and HH2. According to Figure 4b, there are different performances of S185 and HH2—one of the reasons for this could be that HH2 measure the leaves at the lower canopy. However, S185 measured the leaves at the upper canopy.

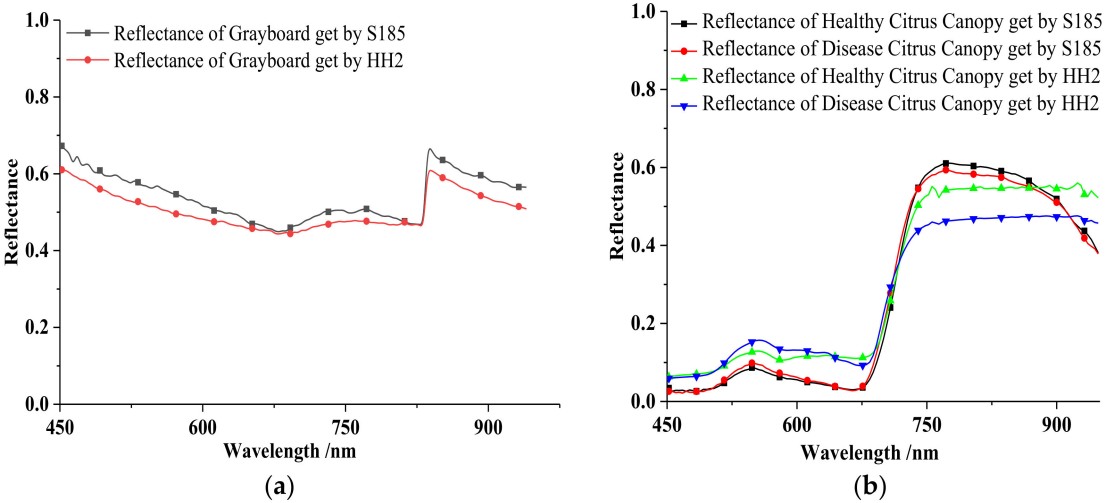

**Figure 4.** *Cont.*

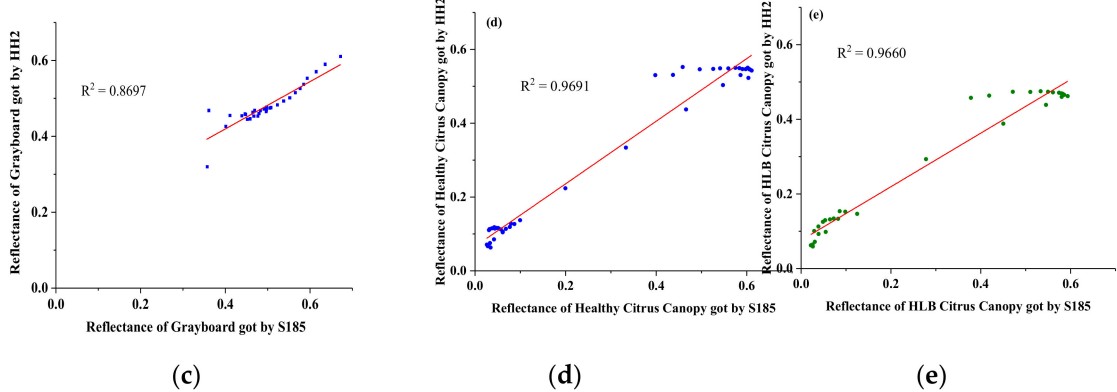

**(c)** **(d)** **(e)**

**Figure 4.** Correlation of data between S185 and HH2. (**a**) Grayboard spectrum curve of S185 and HH2; (**b**) the spectrum of the citrus canopy acquired by S185 and HH2; (**c**) correlation between grayboard spectrum acquired by S185 and HH2; (**d**) correlation between healthy citrus spectrum acquired by S185 and HH2; (**e**) correlation between huanglongbing (HLB) citrus spectrum acquired by S185 and HH2.

Figure 4a shows that the two reflectance curves did not coincide, but there was a high degree of similarity in the changing trend of reflectance. Figure 4b illustrates that the hyperspectral reflectance of the citrus canopy obtained by the two devices, exhibiting the same change from 450 nm to 680 nm, increasing first and then decreasing. Furthermore, the characteristic of a 'green peak' was observed near the wavelength of 550 nm and reached the position of a 'red valley' at the same time near the wavelength of 680 nm, the hyperspectral reflectance of the citrus canopy obtained by S185 declined from 760 nm to 950 nm, which was considerably different from that of the canopy obtained by HH2. Figure 4c shows the data correlation $R^2$ of grayboard between two devices was 0.8697. Figure 4d,e show that their correlation $R^2$ of a healthy citrus canopy and HLB citrus canopy was above 0.96.

### 2.3. UAV Spectral Data Pre-Processing Methods

#### 2.3.1. UAV Hyperspectral Image Stitching

The image data acquired by the S185 hyperspectral camera included hyperspectral cubes and gray images, and image stitching could only be performed after data conversion, hyperspectral image fusion, and geographic information extraction. The stitching process is shown in Figure 5. In this study, we used Cube–pilot for data format conversion, ENVI for data fusion, Waypoint–Inertial Explorer for geographic information interpretation, and Agisoft PhotoScan for image stitching.

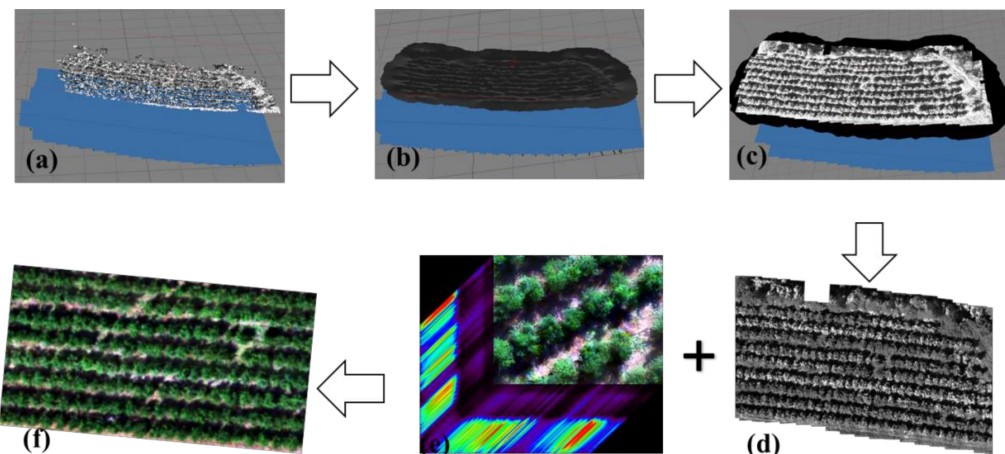

**Figure 5.** UAV hyperspectral image stitching. (**a**) Point cloud generation; (**b**) grid generation; (**c**) texture generation; (**d**) panoramic gray orthophoto; (**e**) hyperspectral image; (**f**) panoramic hyperspectral orthophoto.

### 2.3.2. Canopy Spectral Data Extraction at Pixel Level and Dataset Establishment

As the symptoms of HLB are not always seen in the whole plant, the spectral characteristics of the same canopy are not evenly distributed, and the spectral data of the middle and the edge of the canopy are not consistent. If we take the whole plant as the sample, it will weaken the characteristics of the spectrum samples of the HLB disease; if we set a certain number and size of ROI on the plant canopy and extract the spectrum data of each ROI, we will need a considerably large amount of human and computing resources. Therefore, in order to maximise the diversity of the canopy spectral characteristics, in this paper, we propose a pixel-level canopy spectral data extraction scheme to ensure that the spectral samples of the central and marginal regions of the canopy can be used for training, to ensure the diversity of the model training samples. Through the canopy spectral data extraction at the pixel level, more than 1.2 million samples of the H spectrum were obtained along with more than 1.04 million samples of the HLB spectrum. The obtained sample dataset was divided into the training set and the validation set in the ratio of 3:7.

### 2.3.3. UAV Hyperspectral Data Denoising and Abnormal Sample Removal

Because of factors (such as sensors, environment, and optical effects), noise was inevitably generated. It is extremely important to denoise and smooth hyperspectral data to obtain a better detection performance. Therefore, the *Daubechies wavelet* (*db N*) [36] was adopted in this study for denoising, which is an orthogonal, continuous, and tightly supported wavelet algorithm. The *db* wavelet algorithm involves many parameters, including the selection of the threshold function, the number of decomposition layers, the order of the wavelet function, and the value of the threshold. As N increased, the spectral curve became relatively smooth after wavelet reconstruction. In this study, the denoising effects of the *db4*, *db8*, and *db16* wavelet basis functions were compared. Among them, the spectral curve reconstructed by *db16* was close to the original curve, which could better preserve the characteristics of the spectral curve after the noise removal. Therefore, *db16* was selected as the wavelet basis function for denoising. There are two methods for choosing the threshold function: soft threshold and a hard threshold. The wavelet coefficients of the soft threshold function in processing have better continuity, the signal does not cause additional oscillation, and it has better smoothness, which is more suitable for processing hyperspectral data.

The choice of decomposition layers is closely related to the strength of the noise. The greater the noise intensity is, the higher degree of decomposition is required, and the higher possibility of distortion is. Considering that the weather on the day of data collection is dry and the light is sufficient, the main noise comes from the device itself, so the number of decomposition layers is three usually.

The threshold is a key parameter of a *db* wavelet, and the choice of threshold has a direct relationship with the denoising effect. The particle swarm optimisation algorithm (PSO) was adopted to determine the optimal threshold for the *db* wavelet. The final denoising process is shown in Figure 6.

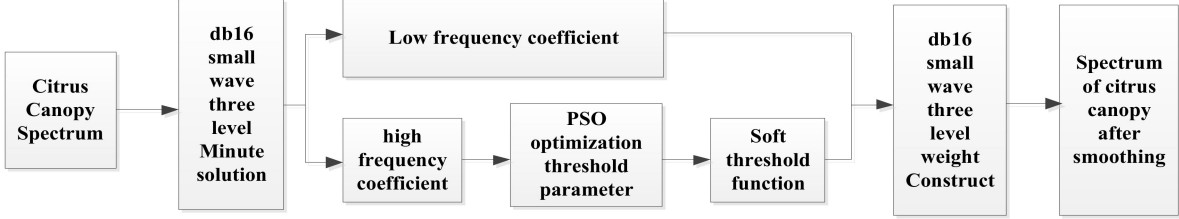

**Figure 6.** Complete denoising process.

After the denoising process, the reconstructed spectral curve fit the original spectral curve, and the curve was highly smooth, retaining the original spectral characteristics. One of the denoising results is shown in Figure 7.

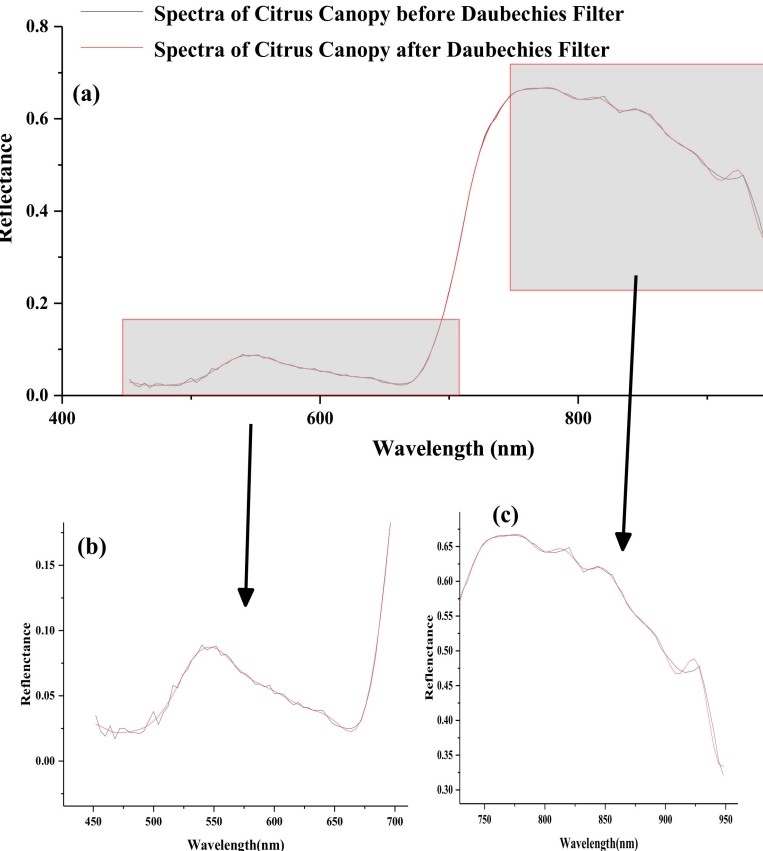

**Figure 7.** Results of smooth denoising: (**a**) Results of full wavelength after smooth denoising; (**b**) results of visible wavelength after smooth denoising; and (**c**) results of near-infrared wavelength after smooth denoising.

In contrast, there were some mixed pixels in the hyperspectral data. To further exclude the disturbed pixel samples and some abnormal samples caused by the interference of environmental factors, isolation forest [37] was adopted, which is an unsupervised anomaly sample detection method suitable for continuous data and is robust to the detection of high-dimensional data, such as the data case in this study. The isolated forest algorithm was implemented with the Python language. There were three main parameters involved, namely, the number of trees, the number of samples per tree, and the number of features extracted by training. The data dimension in this study was 125, and the amount of data was relatively large. Referring to empirical practices, only 64 bands were selected as the features for each training, the number of trees was set to 100, and the number of tree samples was 128 for each training. The test results are shown in Figure 8.

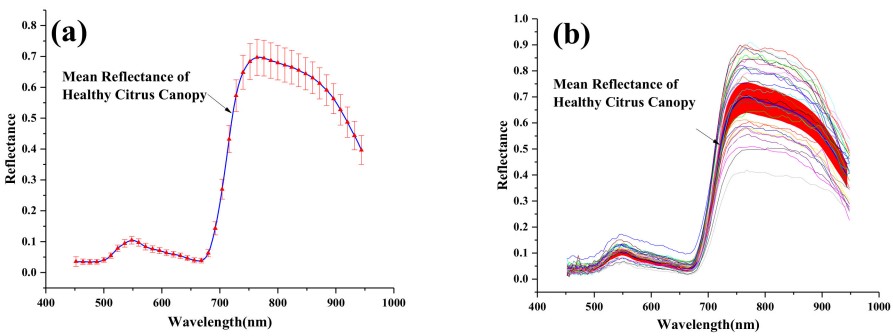

**Figure 8.** *Cont.*

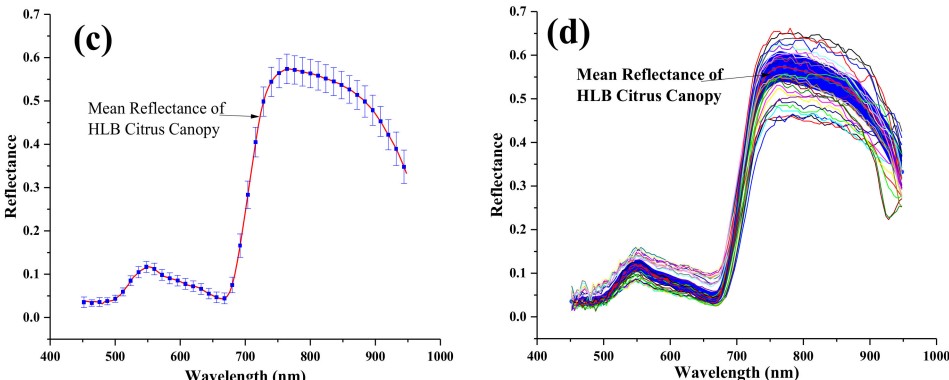

**Figure 8.** Result of abnormal samples test. (**a**) The normal range of canopy spectrum of healthy citrus plants; (**b**) the abnormal spectrum of the canopy in healthy citrus plants; (**c**) the normal range of the canopy spectrum of HLB citrus plants; (**d**) the abnormal spectrum of the canopy in HLB citrus plants.

## 3. Method Description

### 3.1. Spectral Transformation

A mathematical transformation of the spectral curve is helpful to strengthen the spectral features, to identify features artificially, and to build more generalised models. Spectral differential processing is beneficial to eliminate the effects of background, soil, and atmospheric scattering [38]. In this study, the first-order derivative reflectance spectra (FDR) and the inverse logarithmic reflectance spectra (ILR) were compared. The comparison results revealed that FDR could effectively reduce the influence of a part of the linear background and noise in the spectral information on the ground target spectrum, which was more conducive to the data analysis. Because the actual sampling process of the spectrum was discrete, FDR was calculated by using the difference calculation method.

Figure 9 shows the comparison of the original spectrum and the FDR spectra. Figure 9a clearly shows that the 'green peak' characteristic reflectance of the HLB-infected plants was higher than that of the H plants. The steep slopes appearing near the red edge band were higher and steeper. In the vicinity of near-infrared, the canopy spectral reflectance of the H plants was higher than that of the HLB citrus canopy. An analysis of the main reason for this difference revealed that the HLB plants were subject to disease stress, internal physiological structure changes, reduced chlorophyll content, weakened photosynthesis, and reduced ability to absorb water and nutrients. Thus, these plants exhibited external symptoms, such as mottling and yellowing.

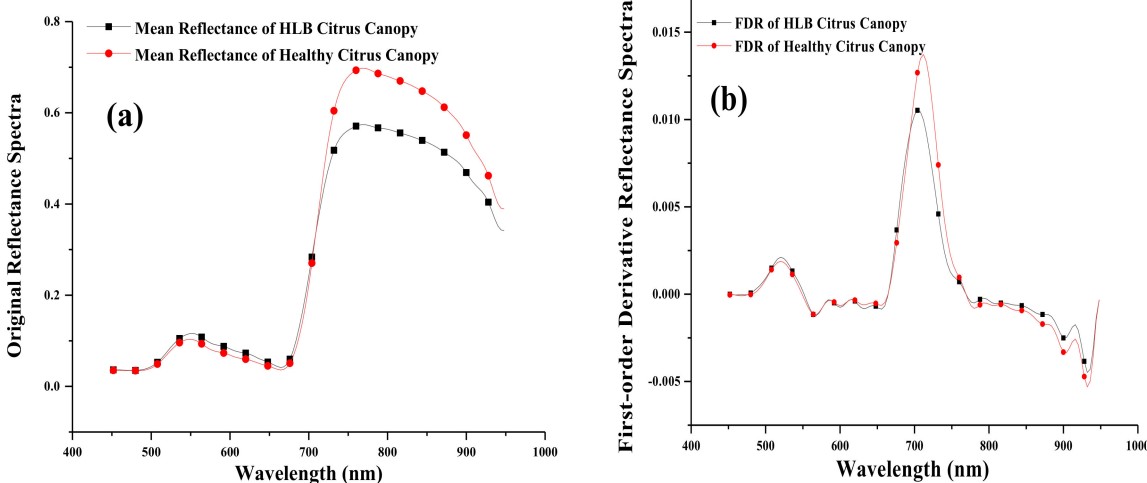

**Figure 9.** Original spectrum and first-order derivative reflectance spectra (FDR). (**a**) Original reflectance spectra; (**b**) FDR reflectance spectra.

As shown in Figure 9b, FDR made the spectral characteristics of the citrus canopy more obvious. The characteristic peaks appeared near 525 nm and 720 nm. The characteristic peaks of HLB-FDR and Healthy-FDR near 525 nm reached the peak at the same time, and the peak of HLB-FDR at this position was higher. For the characteristic peak near 720 nm, HLB-FDR reached the peak first, and the peak was lower than the peak of the H plant canopy. The absolute value of the FDR value in the near-infrared band was larger for Healthy-FDR than for HLB-FDR. This implied that the canopy spectrum attenuation of healthy plants in the near-infrared band was more intense.

## 3.2. Canopy Spectral Characteristic Parameter

The hyperspectral data extracted through the canopy carried a considerable amount of the citrus canopy spectral information, which could reflect the physiological characteristics of the canopy. In the visible wavelength range, the maximum derivative in the different chromatographic bands is called the 'edge' of the chromatogram. In the case of hyperspectral data, the positions of 'blue edge', 'yellow edge', and 'red edge' are the most commonly used characteristic parameters. In the study of the vegetation spectra, 'green peaks' and 'red valleys' are closely related to the physiology of vegetation. The 'green peak' feature of vegetation is closely related to the chlorophyll content of vegetation. Vegetation has a strong ability to absorb red light, thus producing the 'red valley' characteristic. The 'green peak reflection height' and 'red valley absorption' are used to reflect the light absorption capacity of the 'green peak' and 'red valley' wavelength positions, respectively. In the near-infrared band, the red edge is a unique spectral characteristic of green vegetation, such as the position of the red edge $\lambda_R$, the red edge amplitude $D_R$, and the red edge area $SD_R$. Table 2 shows the canopy spectral characteristic parameter information adopted as the features in this study.

**Table 2.** Canopy spectral characteristic parameter information.

| Spectral Characteristics | Abbreviations/Formula | Definition |
| --- | --- | --- |
| Blue Edge Amplitude | $D_B$ | Maximum value of FDR in the wavelength range of 490–470 nm |
| Blue Edge Position | $\lambda_B$ | Wavelength corresponding to the maximum FDR in the 490–470 nm range |
| Blue Area | $SD_B$ | Integration of FDR in the wavelength range of 490–470 nm |
| Yellow Edge Amplitude | $D_Y$ | Maximum value of FDR in the wavelength range of 560–620 nm |
| Yellow Edge Position | $\lambda_Y$ | Wavelength corresponding to the maximum FDR in the range of 560–620 nm |
| Yellow Area | $SD_Y$ | Integration of FDR in the wavelength range of 560–620 nm |
| Red Edge Amplitude | $D_R$ | Maximum value of FDR in the wavelength range of 640–780 nm |
| Red Edge Position | $\lambda_R$ | Wavelength corresponding to the maximum FDR in the range of 640–780 nm |
| Red Area | $SD_R$ | Integration of FDR in the wavelength range of 640–780 nm |
| Green Peak Value | $P_G$ | Maximum value of spectral reflectance in the wavelength range of 510–570 nm |
| Green Peak Position | $\lambda_R$ | Wavelength corresponding to the maximum FDR in the range of 510–570 nm |
| Green Peak Reflection Height | $1 - \dfrac{R_S + \frac{R_E - R_S}{\lambda_E - \lambda_S} * (\lambda_E - \lambda_S)}{R_C}$ | Maximum intensity of the spectral reflectance in the wavelength range of 510–570 nm |
| Red Valley Value | $P_{R0}$ | Minimum spectral reflectance in the 640–700 nm wavelength range |
| Red Valley Position | $\lambda_{R0}$ | Wavelength corresponding to the minimum value of spectral reflectance in the range of 640–700 nm |
| Red Valley Absorption | $1 - \dfrac{R_C}{R_S + \frac{R_E - R_S}{\lambda_E - \lambda_S} * (\lambda_E - \lambda_S)}$ | Absorption intensity of the minimum spectral reflectance in the wavelength range of 510–570 nm |

Note: $R_C$, $R_S$, and $R_E$ denote the spectral reflectance at the centre, start, and end points of the absorption feature; $\lambda_S$ and $\lambda_E$ represent the wavelength at the beginning and the end of the reflection feature, respectively.

### 3.3. Band Selection Based on Genetic Algorithm (GA) with Improved Selection Operator

Band selection is performed to select a subset of bands that play a major role in hyperspectral images, to considerably reduce the data dimension of hyperspectral images and retain useful information more completely. In this study, the genetic algorithm (GA), which is an evolutionary algorithm for stochastic global search, which is often used to optimise the characteristic band of hyperspectral data [39], was adopted. The realisation of the citrus canopy spectral feature band selection based on GA was performed in the Python environment. The core scheme of the algorithm is as follows:

(1) Binary coding was used; (2) RMSE was set as the adaptive function; (3) the contribution degree of the bands for modelling was used as the sorting basis. Herein, individuals with higher contributions were more likely to be selected; (4) single-point cross was set as the cross-recombination method; (5) uniform mutation was set as the mutation method, which could effectively increase the diversity of the group; and (6) the number of iterations for the termination condition was selected, and the number of genetic iterations was set to 1500. After the iterations, the algorithm eventually converged on the most suitable individual in the environment, which was the selected band combination.

### 3.4. Vegetation Index Features Constructed on the Basis of Feature Bands

The vegetation index (VI) is a sign considered to reflect the relative abundance and activity of green vegetation (dimensionless) [40]. Through the linear or nonlinear combination of reflectivity between different remote sensing bands, it is possible to effectively synthesise the related spectral signals, enhance a certain feature or detail of vegetation, and reduce the non-vegetation information. It is a comprehensive expression of the leaf area index, green biomass, chlorophyll content, coverage, and absorbed photo-synthetically active radiation of green vegetation. In this study, Vis (that are often used in green vegetation research) were selected for the feature extraction. The VI constructed by the feature bands extracted by GA, and shown in Table 3, represented the characteristics of HLB.

**Table 3.** Vegetation index information.

| Name | Formula | |
|---|---|---|
| Ratio Vegetation Index (RVI) | $\mathrm{RVI} = \frac{\rho_{NIR}}{\rho_{RED}}$ | Jordan, 1969 [41] |
| Difference Vegetation Index (DVI) | $\mathrm{DVI} = NIR - R$ | Matsas, 1992 [42] |
| Normalised Vegetation Index (NDVI) | $\mathrm{NDVI} = \frac{\rho_{NIR} - \rho_{RED}}{\rho_{NIR} + \rho_{RED}}$ | Clark, 1973 [43] |
| Enhanced Vegetation Index (EVI) | $\mathrm{EVI} = 2.5 \times \frac{\rho_{NIR} - \rho_{RED}}{\rho_{NIR} + 6.0\rho_{RED} - 7.5\rho_{BLUE} + 1}$ | Vlassara, 1995 [44] |
| Triangle Vegetation Index (TVI) | $\mathrm{TVI} = 0.5 \left[ 120 \left( \rho_{NIR} - \rho_{GREEN} \right) \right. $ $\left. -200 \left( \rho_{RED} - \rho_{GREEN} \right) \right]$ | Borge, 2001 [45] |
| Normalised Greenness Vegetation Index (NDGI) | $\mathrm{NDGI} = \frac{\rho_{GREEN} - \rho_{RED}}{\rho_{GREEN} + \rho_{RED}}$ | Lichtenthaler, 1996 [46] |
| Green Ratio Vegetation Index (GRVI) | $\mathrm{GRVI} = \frac{\rho_{NIR}}{\rho_{GREEN}}$ | Anatoly, 1996 [47] |
| Chlorophyll Vegetation Index (CVI) | $\mathrm{CVI} = \rho_{NIR} \frac{\rho_{RED}}{\rho_{GREEN}^2}$ | Vincini, 2008 [48] |

### 3.5. Modelling Methods

In this study, a support vector machine (SVM) [49] was used to establish the citrus canopy extraction model, and the neural network model was used to construct the HLB detection model. The stacked autoencoder (SAE) neural network is a neural network model formed by stacking multiple layers of the sparse autoencoder [50]. The full connection (FC) neural network is one of the most basic neural network models and can handle one-dimensional vectors similar to hyperspectral data well. Therefore, in this study, we selected FC as the classifier to train the HLB detection model.

The SAE neural network is an algorithm that classifies the encoding results after encoding and compressing the data. In this study, the 125-dimensional hyperspectral data were encoded three times to obtain 16-dimensional features. The 16-dimensional features were then classified with an FC neural network to establish an HLB detection model.

To prevent over-fitting, the dropout layer was introduced to the network and set to 0.5. ReLu was selected for the activation function in the hidden layer, and sigmoid for the classification layer was used. Cross entropy (CE) was chosen as the loss function. The back-propagation operation was executed on the basis of the gradient descent principle. A callback function (ReduceLRonPlateau) was used to automatically adjust the learning rate.

Different data sources as the training set in the same network have different modelling performances. In this study, a full-band original spectrum and the FDR spectra, characteristic band, VIs, and multi-source features were input separately into the SAE neural network for the sake of comparison. In Figure 10a, the FC neural network was used. The feature bands extracted by GA contained a total of nine sub-bands in the blue, green, and red band region. Considering that vegetation often has obvious characteristics in the near-infrared band, the algorithm results of the iterative GA searched back and found that the near-infrared band with a centre wavelength of 852 nm had a high frequency. Therefore, this band was also used as the characteristic band, so there were 10 characteristic bands selected by the GA input to the network, shown in Figure 10a.

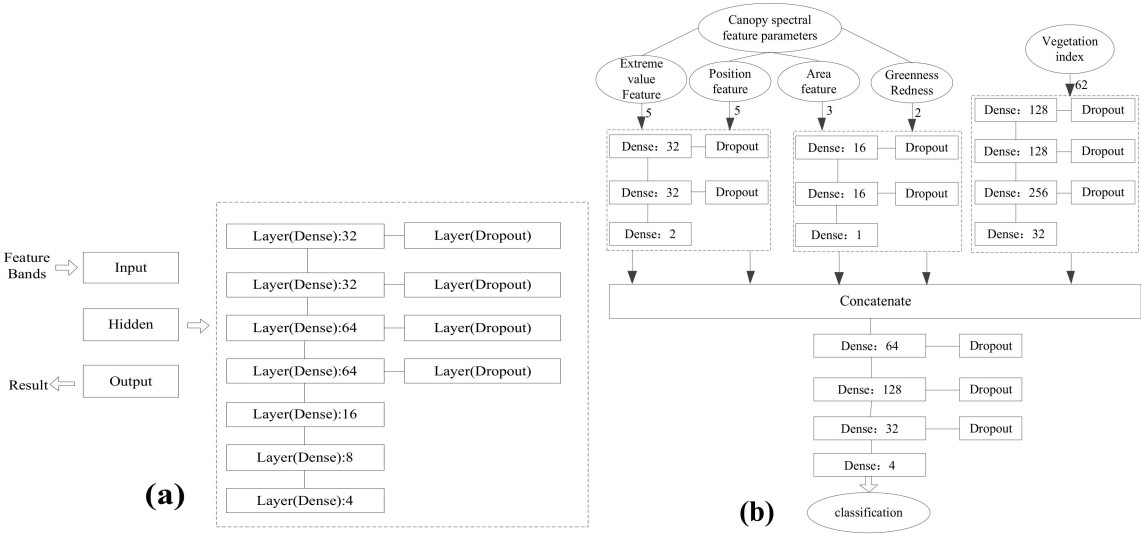

**Figure 10.** (**a**) Network structure of HLB detection model based on the characteristic band; (**b**) the network structure of HLB detection model based on multi-feature input.

In Figure 10b, multiple features were input to the FC neural network, including 62 VIs and 15 canopy feature parameters. The 62 VIs were constructed according to Table 4, where the vegetation indices, including RVI, DVI, NDVI, EVI, TVI, NDGI, GRVI, and CVI, were calculated on the basis of the characteristic band selected by GA. Therefore, the red and green light bands with multiple feature bands were sequentially substituted into the vegetation index model for calculation.

**Table 4.** The number of features built from vegetation indices.

| Vegetation Index | Number of Features |
|---|---|
| RVI, DVI, NDVI, EVI | 3 |
| GRVI | 5 |
| TVI, NDGI, CVI | 15 |

Furthermore, 15 spectral characteristic parameters were extracted from the citrus canopy, including the extreme value features (peak or valley), location features, area features, and greenness and redness features. The FC neural network input and processed five types of features at the same time through five input layers, as shown in Figure 10b, and established a multi-feature fusion HLB detection model. The model, shown in Figure 10b, was composed of seven parts, namely, the extreme value feature

processing layer, the position feature processing layer, the area feature processing layer, the greenness and redness feature processing layer, the VI feature processing layer, and the feature stitching layer and classification layer.

## 4. Experiments and Results

### 4.1. Results of Feature Band Extraction

Feature band extraction was performed on the basis of the contribution to the accuracy of the model. The contribution was equal to the accuracy rate minus 0.01*band number. In the case of the same number of bands, the higher the accuracy of the individual as compared to the model was, the higher was its contribution. For the same model accuracy rate, the smaller the number of individual genes was, the higher was the contribution value. In GA, cross probability and mutation probability have a certain effect on the final output of the model. If the cross probability is set too large, it will destroy the goodness of the population, deteriorate the selected band result, and reduce the accuracy of the model. A very small cross probability affects the evolution of the algorithm and deteriorates the convergence performance. When the cross probability was set to 0.5, and the mutation probability was set to 0.02, after 1500 iterations, the output bands obtained by GA were 468 nm, 504 nm, 512 nm, 516 nm, 528 nm, 536 nm, 632 nm, 680 nm, and 688 nm. The accuracy of the model obtained by this band combination was 91.92%. The wavelength range had one band at the blue wavelength, five bands at the green wavelength, and three bands at the red wavelength. The results of feature band extraction based on GA were shown in Table 5.

**Table 5.** Result of genetic algorithm.

| Parameter | Central Wavelength of the Selected Band (Nm) | Accuracy |
|---|---|---|
| Cross probability = 0.5 Mutation probability = 0.02 | 468, 504, 512, 516, 528, 536, 632, 680, 688 | 91.92% |

### 4.2. Modelling Results

#### 4.2.1. SAE Modelling for HLB Detection Based on Full-Band Original and FDR Spectra

The effect of SAE modelling on the full-band original spectrum is shown in Figure 11a–c. In Figure 11a, the curve converged when iterating to 46 times, the loss of the training set was 0.587, and the loss of the validation set was 0.682. In Figure 11b,c, the original SAE detection model converged after seven iterations, the classification accuracy of the training set was 72.76%, the loss was 0.5475, the classification accuracy of the validation set was 74.87%, and the loss was 0.5224. The SAE modelling effect of the full-band FDR is shown in Figure 11d–f. In Figure 11d, it converged when iterating to 31 times, the loss of the training set was 0.891, and the loss of the validation set was 0.904. In Figure 11e,f, the classification accuracy of the training set was 89.88%, and the loss was 0.2864 after 16 iterations of the derivative spectrum. The classification accuracy of the validation set was 90.01%, and the loss was 0.2537.

We found that the loss value of FDR during reconstruction was larger than the original spectrum, but the classification effect was better, and the loss value of the classification model was lower. This phenomenon might be attributed to the fact that FDR did not show obvious rules with increasing wavelength and the value of FDR fluctuated considerably. The reconstructed spectrum based on a nonlinear transformation could not restore the spectrum well. Therefore, the loss of FDR in the SAE network reconstruction process was relatively large. However, because the characteristics of FDR were more obvious, the accuracy of detection could be considerably improved, the detection error of the model was reduced, and the loss value decreased.

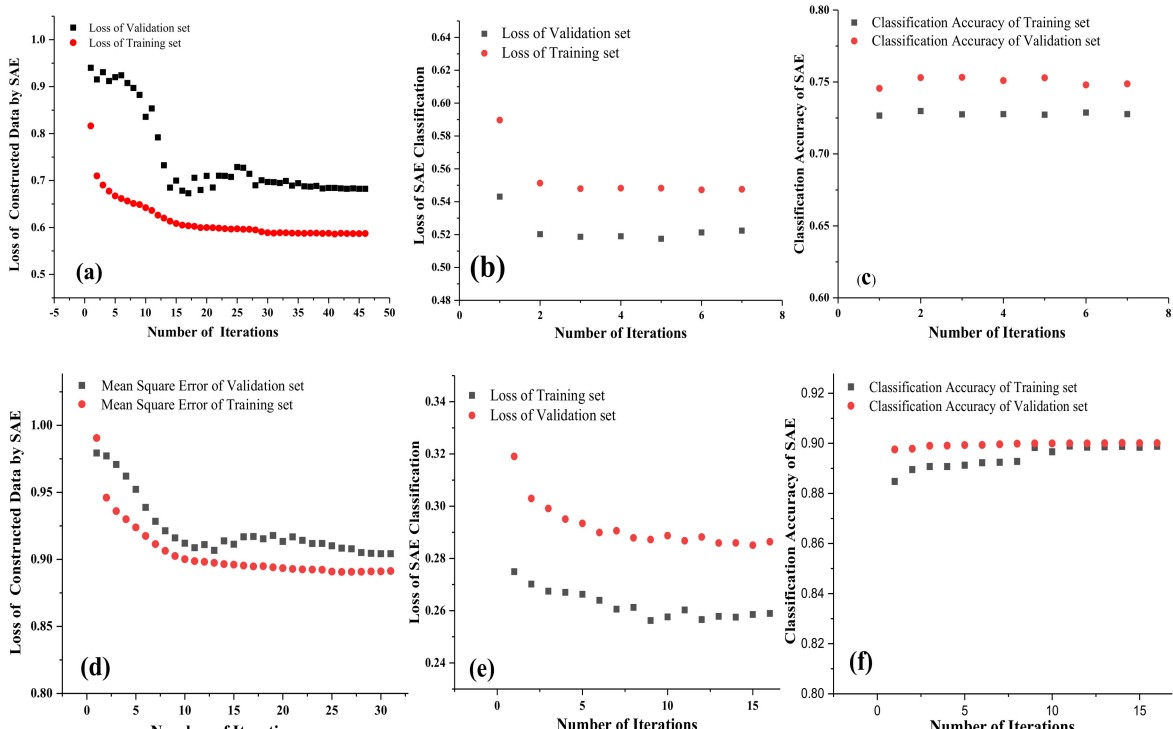

**Figure 11.** Results of stacked autoencoder (SAE) neural network; (**a**) loss of the original reconstructed spectrum by SAE; (**b**) loss of SAE classification; (**c**) SAE classification accuracy; (**d**) loss of constructed FDR by SAE; (**e**) loss of HLB detection model by FDR-SAE; (**f**) accuracy of HLB detection model by FDR-SAE.

### 4.2.2. HLB Detection Model Based on Feature Band

With the characteristic band as the model input, after 70 iterations, the classification results obtained are shown in Figure 12a,b. The classification accuracy of the training set was 96.01%, and the loss was 0.0992. The classification accuracy of the validation set was 97.16%, and the loss was 0.076. Compared with the full-band case, the accuracy of HLB detection by this model was significantly improved. The loss value was also significantly reduced, and the parts of the model that needed to be adjusted were reduced. We observed that although the neural network could automatically extract the data features, this feature was not necessarily suitable for practical purposes. If the extracted features were targeted, the model effect could be better optimised.

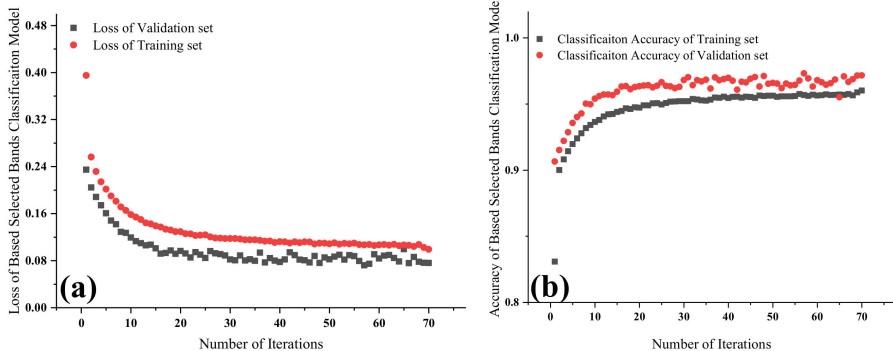

**Figure 12.** Results of HLB detection model-based feature bands; (**a**) loss of HLB detection model based feature bands; (**b**) accuracy of HLB detection model based feature bands.

### 4.2.3. Results of HLB Detection Model Based on Multi-Feature Fusion

Although the HLB detection model based on the characteristic band achieved good results, the model did not establish a relationship with the vegetation physiology, and it was difficult to explain the high accuracy achieved by the model. Figure 13 shows the performance of HLB detection based on multi-source features which were input to the SAE neural network, including 62 VIs and 15 canopy feature parameters. After 87 operation iterations, the classification accuracy of the training set was 99.33%, and the loss was 0.02662. The classification accuracy of the validation set was 99.72%, and the loss was 0.0119. Because of the strong characteristics of the input data, the discriminative ability was improved, and the robustness of the model was guaranteed. Thus, we confirmed that the combination of the physiological characteristics of vegetation considerably enhanced the establishment of the HLB detection model. Regrettably, the complexity of the structure of the model was also considerably increased.

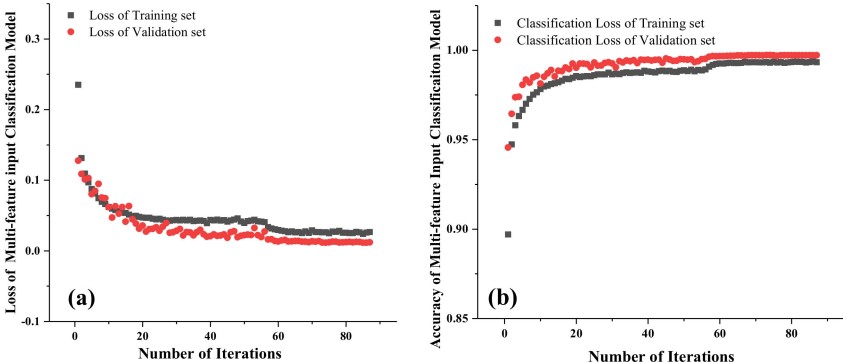

**Figure 13.** Results of HLB detection model based on multi-feature fusion; (**a**) loss of multi-feature input classification model; (**b**) accuracy of multi-feature input classification model.

### 4.2.4. Verification of Model Detection Effect

To verify the feasibility of the established model, hyperspectral remote sensing detection in the test area was carried out. A multi-feature fusion HLB detection model based on the SAE neural network was built in the pixel level of the UAV hyperspectral images. A citrus canopy label was established by using a quadratic kernel SVM in ENVI. The dataset used to establish the canopy label obtained the first three components after MNF processing. The hyperspectral data were read and detected pixel by pixel on the basis of the extracted citrus canopy label, and the detection results were output at the pixel level, as shown in Figure 14. All of the diseased trees were detected in the experiment; however, eight among the 293 healthy plants were incorrectly recognised as HLB.

A 'circle' in Figure 14 indicates that the disease tree detection was correct, and a box indicates a misjudgement of ground-truth data. The details of the testing results for the citrus canopy are shown in Figure 15.

Figure 15a shows the 'flower dots' in the misjudgement result for healthy plants. In this case, some pixels in the healthy plant were determined to HLB-infected ones which are named flower dots. There might be two reasons for the 'flower dots'. The first is that mixed pixels were generated by light, ground, and reflected light from other places. The second might be the incorrect labelling, as HLB-infected plants do not always show HLB symptoms everywhere in the canopy, which encouraged the system to include the health spectrum of the canopy of infected plants into the model for training. Figure 15b,c shows the D1 and D2 level detecting result with distribution of the HLB infection in the canopy. The extent of disease can be calculated according to the proportion of the HLB-infected distribution in a plant. The testing results revealed that the coverage of HLB pixels in the D1 and D2 disease trees was consistent with the definition of the disease level: The HLB coverage of the D1 level was less than 50% and that of the D2 level was more than 50%.

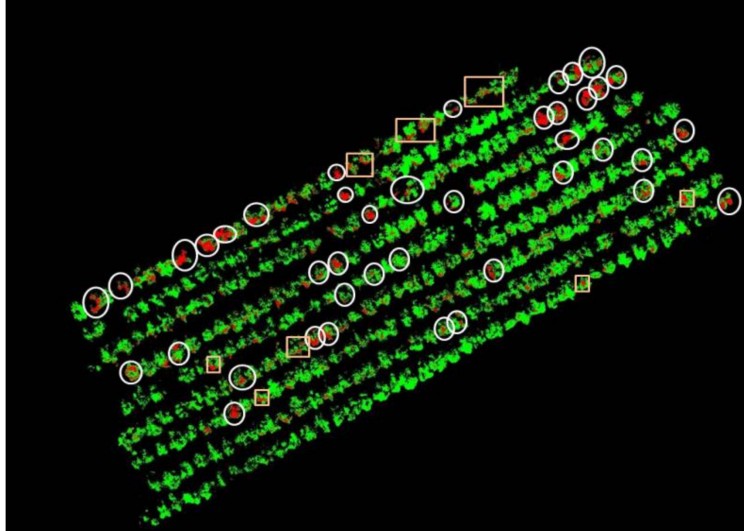

**Figure 14.** Testing results of the test area.

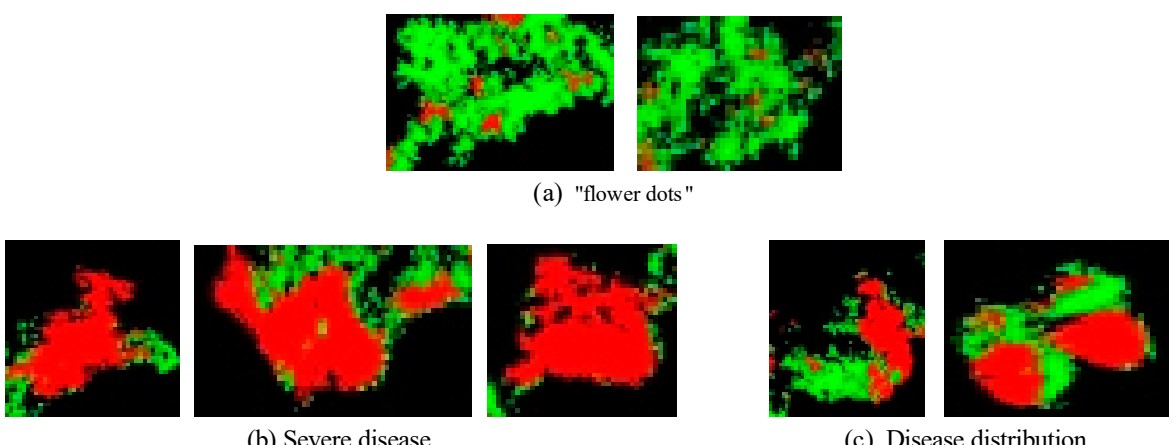

**Figure 15.** Details of testing results of the citrus canopy.

## 5. Discussion

Although the above scheme can achieve relatively intelligent HLB detection in large citrus orchards, it still needs to be discussed with the existing research to better reflect the advantages of this scheme.

Reference [33] proposed a new algorithm of 'extended spectral angle mapping (ESAM)' to classify HLB for airborne hyperspectral and satellite multispectral remote sensing. It is found that the proposed ESAM method can distinguish HLB better than Mahalanobis distance and K-means, and the classification accuracy rate reaches 86%. Reference [33] studied the extraction of pure endmember and red-edge position of HLB, using the extracted features to help classify HLB. The model used in this paper is FC. Compared with the traditional machine learning model, it has a more powerful classification ability for large amounts of data, and can handle high-dimensional data and data with unobvious features. In addition, this paper also studies the fusion of multiple commonly used spectral features to strengthen the model's ability to recognise HLB features and improve the model's anti-interference ability.

It can be seen that Reference [34] uses the UAV hyperspectral remote sensing method to detect HLB, uses the Quadratic SVM to establish a discriminant model for the derivative spectrum of 270 bands, and the classification accuracy rate reaches 94.7%. However, the data used in this model is a sample of ROI randomly extracted from the citrus canopy. This data set has strong randomness, resulting

in poor model stability. Moreover, it takes a lot of time to extract ROI manually. Reference [34] used the classification results of ROI samples to determine whether citrus was infected with HLB. However, this paper extracts spectral samples based on a single pixel, which increases the richness and anti-interference ability of the samples. In addition, this paper diagnoses HLB based on the pixel level, and uses the multi-feature fusion FC model to establish the classification model, which can more intuitively obtain the distribution of diseases in the canopy, and provide guidance for cutting off the branches and stems infected with HLB. It can be said that this paper is a further improvement of Reference [34].

## 6. Conclusions

In this study, we conducted HLB detection at the low altitude scale based on UAV hyperspectral remote sensing and developed a complete data processing and analysis scheme. The following conclusions were drawn:

(1) The correlation coefficients $R^2$ of the hyperspectral data collected by HH2 and S185 exceeded 0.96, which implied that the UAV hyperspectral data captured by S185 were consistent with the ground hyperspectral data.

(2) The Daubechies wavelet algorithm combined with PSO had a considerable smoothening and denoising effect on the hyperspectral data. The isolated forest was effective in removing the abnormal samples from the dataset.

(3) The improved GA provided the following results of feature band extraction for HLB detection—468 nm, 504 nm, 512 nm, 516 nm, 528 nm, 536 nm, 632 nm, 680 nm, 688 nm, and 852 nm.

(4) The SAE neural network based on multi-feature fusion, including 62 VIs and 15 spectral characteristic parameters of the citrus canopy, achieved a high performance on HLB detection, and the model exhibited a classification accuracy of 99.33% for the training set and 99.72% for the verification set.

(5) The pixel-level hyperspectral remote sensing datasets avoided the impact of the inconsistency between the canopy centre and the canopy edge reflectance intensity, and improved the diversity of the dataset and the robustness of the model, but the shortcoming was that the healthy plants were prone to being misjudged as HLB.

**Author Contributions:** Conceptualisation, X.D. and Z.Z. (Zihao Zhu); methodology, Z.Z. (Zihao Zhu) and X.D.; software, Z.Z. (Zihao Zhu); validation, Z.Z. (Zihao Zhu), X.D., and J.Y.; investigation, X.D., Z.Z. (Zihao Zhu), J.Y., X.Y., Z.Z. (Zheng Zheng), Z.H., and S.W.; resources, Y.L.; data curation, X.D., Z.Z. (Zihao Zhu), J.Y.; writing—original draft preparation, X.D. and Z.Z. (Zihao Zhu); writing—review and editing, Y.L.; visualisation, Z.Z. (Zihao Zhu) and J.Y.; supervision, X.D. and Y.L.; project administration, Y.L.; funding acquisition, Y.L. All the authors have read and agreed to the published version of the manuscript.

**Funding:** This research work was supported by the Key-Area Research and Development Program of Guangdong Province (Grant No. 2019B020214003), the National Natural Science Foundation of China (Grant No. 61675003), and the Key-Areas of Artificial Intelligence in General Colleges and Universities of Guangdong Province (Grant No. 2019KZDZX1012), University Student Innovation Cultivation Program of Guangdong, China (Grant No. pdjh2019b0079).

**Acknowledgments:** The authors would like to acknowledge Guangzhou Xingbo Company for the contributions in the UAV data collecting phase. We also gratefully acknowledge the support of the Huanglongbing laboratory, South China Agricultural University.

**Conflicts of Interest:** The authors declare no conflict of interest.

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
