# Peer review of "Detection of Citrus Huanglongbing Based on Multi-Input Neural Network Model of UAV Hyperspectral Remote Sensing"

_remotesensing, doi:10.3390/rs12172678_

Round 1

Reviewer 1 Report

This is an overall good manuscript describing the use of UAV hyperspectral imagery to identify citrus trees which are affected by HLB.

Throughout the manuscript there are some minor English issues (a lot of misuse of indefinite articles) that if fixed would improve readability.

In L155 - how were the spectra combined? Simple averaging, or was a response function (FWHM or other) used to model the response?

L217: what is the measure of quality of fit/noise removal? How was this assessed?

section 2.3 is methods, not data

There are a couple sections where the text is unclear. This includes L225, and all of the section L243-260 which should be looked at and edited/reworded. 

Throughout the text, the words "whitebroad" and "graybroad" are used instead of the correct whiteboard and grayboard. In addition, it was unclear if both were used, or only a grayboard. What % reflectance grayboard was used is also never defined. What was the grayboard material?

Figure 4 needs several improvements - the axis labels and figure sub-labels need to be fixed. For 4c,d, and e - what wavelength is this for? Or is it an average of the entire spectra? Use consistant X and Y axes!

What is the issue with the grayboard spectra in 4a? I don't know any commonly used standard whiteboard/grayboard that had such a sharp spectral feature in it?

Figure 8 - use the same y axis across all the sub-figures!

Table 2, middle column - these are almost all abbreviations, not formulas, What was the basis for choosing these specific canopy spectral features vs other canopy spectral features?

Figure 10 is cropped and not all visible

L360-366 - Why were multiple values for B, G and R used as inputs? This leads to very highly correlated datasets in your VIs - tough you have 62 VIs, they is massive amount of information overlap between them! you reduce the dimensionality in 3.3, only to end up feeding the data into correlated indexes.

Author Response

Reviewer 1

This is an overall good manuscript describing the use of UAV hyperspectral imagery to identify citrus trees which are affected by HLB.

Throughout the manuscript there are some minor English issues (a lot of misuse of indefinite articles) that if fixed would improve readability.

In L155 - how were the spectra combined? Simple averaging, or was a response function (FWHM or other) used to model the response?

 Answer: The method of simple averaging was adopted to combine the spetra.

L217: what is the measure of quality of fit/noise removal? How was this assessed?

 Answer: We calculated the average spectrum and set the confidence interval according to the big sample data, the performance of noise removal was measured by whether it is in confidence interval.

section 2.3 is methods, not data

 Answer: The title of section 2 has been renamed to” Materials and data pre-processing methods”, thank you for the suggestion.

There are a couple sections where the text is unclear. This includes L225, and all of the section L243-260 which should be looked at and edited/reworded. 

 Answer: The text of L225 and L243-260 has been edited. Thank you.

Throughout the text, the words "whitebroad" and "graybroad" are used instead of the correct whiteboard and grayboard. In addition, it was unclear if both were used, or only a grayboard. What % reflectance grayboard was used is also never defined. What was the grayboard material?

Answer: The expression about the calibration board has been revised. The reflectance of grayboard is 60%. The material is unknown。

Figure 4 needs several improvements - the axis labels and figure sub-labels need to be fixed. For 4c,d, and e - what wavelength is this for? Or is it an average of the entire spectra? Use consistant X and Y axes!

Answer: the figure sub-labels and axis labels have been fixed. The “wavelength” is for the full-bands of spectra.

What is the issue with the grayboard spectra in 4a? I don't know any commonly used standard whiteboard/grayboard that had such a sharp spectral feature in it?

Answer: The data of Figure 4a was measured in the orchard. Through comparison in the process of experiment, the spectral characteristics of grayboard with 60% reflectance is consistent with the one of 40%.

Figure 8 - use the same y axis across all the sub-figures!

 Answer: Thank you for the comment, the y axis were labeled automatically by the software according to the data for each sub-figure.

Table 2, middle column - these are almost all abbreviations, not formulas, What was the basis for choosing these specific canopy spectral features vs other canopy spectral features?

 Answer: The middle column of the table 2 represents the abbreviations or formulas of the canopy spectral features, which were chosen on the basis of the original data analysis from L291-L304, also the listed canopy spectral features are quite popular 

Figure 10 is cropped and not all visible

 Answer: Figure 10 has been fixed.

L360-366 - Why were multiple values for B, G and R used as inputs? This leads to very highly correlated datasets in your VIs - tough you have 62 VIs, they is massive amount of information overlap between them! you reduce the dimensionality in 3.3, only to end up feeding the data into correlated indexes.

 Answer: The purpose of using multiple values for B,G and R as inputs is that there are 10 characteristic bands extracted by the GA. Although there is information overlap, our previous studies have shown that it is hard to get a perfect classification performance only using 3 bands. Thanks for the comment, we will consider correlation analysis for these features in the future research.

Reviewer 2 Report

  1. Comments to Author:

Dear Ref. No.: remotesensing-875848

Title: Detection of Citrus Huanglongbing Based on 2 Multi-input Neural Network Model of UAV 3 Hyperspectral Remote Sensing

Abstract: Your abstract is fine. Line 16 is a bit strange when you mention the UAV hyperspectral remote sensing method, It is a tool, not a method in my opinion, so I would change method to imagery. I also think you should mention the classifier used on the abstract.

Introduction:

Your introduction is ok, but it needs to be improved a bit. In the first two paragraphs you must mention that HLB is a bacterial disease caused by Candidatus Liberibacter asiaticus (feel free to correct this info in case any difference was observed on your study site). On line 50, you mention that field detection is affect by environmental factors. Add which factors and what is their influence on the text. On the last paragraph what are the low and high accuracies? This paragraph can be expanded improving the discussion on the mentioned papers. I also felt that when you mentioned that there are other UAV studies to detect HLB that did not used physiological characteristics, I was expecting this to be on of your aim, but you mention only detect HLB infection levels. Use this information to also improve this last paragraph.

Materials and Data:

On line 119-120 you mention that infections levels were more than 50% and less than 50%. You must explain why these levels and which other studies were based on these levels as well (a reference would be good).

Figure 1 is quite poor. I can’t see the coordinates, cant see the google maps image and can barely see the study site image. You should also add 3 figures emphasizing healthy, less than 50% and more than 50% contamination on treetops (closer zoom to the reader see the difference on yellowness).

I also couldn’t find the spatial resolution of the UAV image you used.

Methodology:

On the first two paragraphs od this section, I think you must add more references here, including authors that performed similar image transformation methods for similar applications.

One major issue on this section relates Figure 9 and features described on Table 2 (also the modelling features mentioned on line 362-263 - multiple features were input to the FC neural network, including 62 VIs and 15 canopy feature parameters). You capture parameters from the green, blue, yellow, red and red-edge regions… but where are these features for the NIR region? Figure 9a shows that there is so much information beyond the 780 nm (last wavelength of the red-edge features). You mention later that you the genetic algorithm selected the NIR band of 852, but how about NIR valleys/peaks position, height, absorption (all similar parameters of table 2?). You must clearly justify in the text why this region did not have these parameters used (based on literature as well) otherwise you must include these features and model your results again. In my opinion this is one of the major issues to be corrected on the paper.

Results:

This section is ok, with the consideration made before. I also emphasize that the result of the FDR is expected to be better than using the spectral features, since no NIR parameters were calculated. Notice that FDR kind of smooths the differences between healthy and diseased plants on the NIR region and highlights on the red-edge band. Thus, descriptors on the red-edge should provide better results on the classification.

Discussion:

This section must be deleted and written again from the scratch. You don’t summarize findings (some already mentioned before) in this section. This is to compare, contrast, emphasize your results when compared to other literature. Why they are better, which are limitations compared to other studies, what could be improved considering other studies, what advances there are on your methodology, why these features were important, this classifier is better, etc... and etc… . There is not a single reference here and this is the second major issue of this paper. There is no discussion basically.

Conclusion:

On this section if fine to summarize findings in topics, but may need to be adjusts according to discussion changes.

Best regards,

Author Response

Reviewer 2

Dear Ref. No.: remotesensing-875848

Title: Detection of Citrus Huanglongbing Based on 2 Multi-input Neural Network Model of UAV 3 Hyperspectral Remote Sensing

Abstract: Your abstract is fine. Line 16 is a bit strange when you mention the UAV hyperspectral remote sensing method, It is a tool, not a method in my opinion, so I would change method to imagery. I also think you should mention the classifier used on the abstract.

Answer: Thank you for the suggestion, the expression was changed and the SAE classifier was also mentioned in the abstract part.

Introduction:

Your introduction is ok, but it needs to be improved a bit. In the first two paragraphs you must mention that HLB is a bacterial disease caused by Candidatus Liberibacter asiaticus (feel free to correct this info in case any difference was observed on your study site). On line 50, you mention that field detection is affect by environmental factors. Add which factors and what is their influence on the text. On the last paragraph what are the low and high accuracies? This paragraph can be expanded improving the discussion on the mentioned papers. I also felt that when you mentioned that there are other UAV studies to detect HLB that did not used physiological characteristics, I was expecting this to be on of your aim, but you mention only detect HLB infection levels. Use this information to also improve this last paragraph.

Answer: Thanks for the suggestion. The origin of HLB disease was added, the field detection method is mainly affected by subjective factors, so the incorrect expression about  environmental factors was deleted.  In the last paragraph, actually there are very few literature about HLB detection using hyperspectral imagery, the other UAV studies mentioned in this paper using hyperspectral imagery did not use vegetation indices information and spectral parameters, which is the difference from this paper.

Materials and Data:

On line 119-120 you mention that infections levels were more than 50% and less than 50%. You must explain why these levels and which other studies were based on these levels as well (a reference would be good).

Answer: Thanks for the comment. The infection level was defined according the amount of leaves with obvious symptoms of HLB disease and the results of PCR testing which was used to each plants in the experimental plot.

Figure 1 is quite poor. I can’t see the coordinates, cant see the google maps image and can barely see the study site image. You should also add 3 figures emphasizing healthy, less than 50% and more than 50% contamination on treetops (closer zoom to the reader see the difference on yellowness).

I also couldn’t find the spatial resolution of the UAV image you used.

Answer: The spatial resolution of the UAV image is a grayscale image of 1000*1000 and shown in Table1. The panorama in Figure 1 is a visible light panorama obtained by DJI Go 4. About the level of disease, we did get two levels of HLB, however, it is quite hard to discriminate the level in low-altitude by UAV remote sensing. Therefore, in this study, we only focus the discrimination between healthy category and HLB-infected one. We can neglect the difference on yellowness.

Methodology:

On the first two paragraphs od this section, I think you must add more references here, including authors that performed similar image transformation methods for similar applications.

One major issue on this section relates Figure 9 and features described on Table 2 (also the modelling features mentioned on line 362-263 - multiple features were input to the FC neural network, including 62 VIs and 15 canopy feature parameters). You capture parameters from the green, blue, yellow, red and red-edge regions… but where are these features for the NIR region? Figure 9a shows that there is so much information beyond the 780 nm (last wavelength of the red-edge features). You mention later that you the genetic algorithm selected the NIR band of 852, but how about NIR valleys/peaks position, height, absorption (all similar parameters of table 2?). You must clearly justify in the text why this region did not have these parameters used (based on literature as well) otherwise you must include these features and model your results again. In my opinion this is one of the major issues to be corrected on the paper.

 Answer: We only used some common canopy feature parameters in Table 2 where blue, green, red and red edge information was adopted. Besides that, to take advantage of NIR information, we selected the NIR band of 852nm and used the information to calculate the VIs shown in Table 3. Thank you for the comment, we will clearly describe the detail for Table 3.

Results:

This section is ok, with the consideration made before. I also emphasize that the result of the FDR is expected to be better than using the spectral features, since no NIR parameters were calculated. Notice that FDR kind of smooths the differences between healthy and diseased plants on the NIR region and highlights on the red-edge band. Thus, descriptors on the red-edge should provide better results on the classification.
Answer: Thanks for the agreement.

Discussion:

This section must be deleted and written again from the scratch. You don’t summarize findings (some already mentioned before) in this section. This is to compare, contrast, emphasize your results when compared to other literature. Why they are better, which are limitations compared to other studies, what could be improved considering other studies, what advances there are on your methodology, why these features were important, this classifier is better, etc... and etc… . There is not a single reference here and this is the second major issue of this paper. There is no discussion basically.

 Answer: Thanks for the agreement. I will add in the article.

Conclusion:

On this section if fine to summarize findings in topics, but may need to be adjusts according to discussion changes

Answer: Thanks for the agreement.

Reviewer 3 Report

The title of the manuscript (MS) deals with the "Detection of Citrus Huanglongbing Based on Multi-input Neural Network Model of UAV Hyperspectral Remote Sensing". The MS is globally sound, clear, and well written with an adequate structure as a scientific paper demands. However, I have suggested a few modifications that can result in substantial improvement to this manuscript.

In the "Introduction" section, speaking about the development of modern technology and the advantage of using Remote Sensing (RS) technology, the authors should provide several references to substantiate the claim made in paragraph 3 (that is, provide references to other groups who do or have done research in this topic) to make the introduction more substantial.

Also, please increase the size of all figures such as (Fig1, 2, 5, and 15). I cannot see the text in Figures 1, 4, 7, 8, and 10.

Author Response

Reviewer 3

The title of the manuscript (MS) deals with the "Detection of Citrus Huanglongbing Based on Multi-input Neural Network Model of UAV Hyperspectral Remote Sensing". The MS is globally sound, clear, and well written with an adequate structure as a scientific paper demands. However, I have suggested a few modifications that can result in substantial improvement to this manuscript.

In the "Introduction" section, speaking about the development of modern technology and the advantage of using Remote Sensing (RS) technology, the authors should provide several references to substantiate the claim made in paragraph 3 (that is, provide references to other groups who do or have done research in this topic) to make the introduction more substantial.

Also, please increase the size of all figures such as (Fig1, 2, 5, and 15). I cannot see the text in Figures 1, 4, 7, 8, and 10.

Answer: Thanks for the agreement and suggestion. In the “Introduction” section. We listed several references about detection of crop diseases and insect pests using UAV remote sensing such as references [13-20] and [30-35], which were cited separately in paragraph 5 and 7.  The size of all figures was adjusted. Thank you very much for the comments.

Reviewer 4 Report

According to topic and methodological point of view, this paper follows the fashion. However, it is not well organized. The understanding of agriculture and disease is lack. Further, the part of remote sensing technique also not well described and discussed.

Line17-18: In abstract, please mainly use instrument name, instead of product name.

Line 108 and Title: Please clearly define your purpose. Detection? Or Evaluation of disease degree?

Line107:  Please change ‘the current study’ to ‘this study’.

Line109: What do you mean about “a large-scale area”. Canopy scale? Farm-land scale? Continental scale?

Line117~120:  Please show some picture of sample D1 and D2 leaves. And how many trees were infected? Please explain the infected concentration and method. Further, please represent the number of total, infected, D1, and D2 trees as some Table.

Figure 1: How many times measured? Please represent your measurement schedule as some Table. And, this is very important one. Please show the time and weather condition of UAV measurement. According to measurement time, particularly in solar radiation and air temperature conditions, the performance of canopy spectral reflectance will be changed.

Line127-136: Did you use Pix4D? or another SW? And, did you use only one grey board? How about black, white, and another grey board?

Figure 2: ‘System’? What was the system?

Figures 2 and 4: According to your picture, HH2 might measure the leaves at lower canopy. However, S185 measured the leaves at upper canopy. It may be one of the reasons on the different performance of S185 and HH2 in Figure 4(first_b).

Figure 4(second_b): What is ‘wirtebroad’?

Figure 4(d,e): Which band was used for reflectance?

Figure 7: Please explain why smoothing was used. After smoothing, the spectral plot in the graph will be beautiful, but it will make more error.

Line248: Please write the library or module of python.

Line252: Sample number is enough for training? Please explain that. And did you use only S185 data? How about HH2?

3.2 and 3.4 : I don’t understand why you write these sections in this paper. It is not review paper. I cannot find any relations between your results and these two sections. Please explain that.

Author Response

Reviewer 4

Comments and Suggestions for Authors

According to topic and methodological point of view, this paper follows the fashion. However, it is not well organized. The understanding of agriculture and disease is lack. Further, the part of remote sensing technique also not well described and discussed.

Answer: Thank you for the comment, we will seriously revise the manuscript according to the following suggestions in detail.

Line17-18: In abstract, please mainly use instrument name, instead of product name.

 Answer: the instrument names were added.

Line 108 and Title: Please clearly define your purpose. Detection? Or Evaluation of disease degree?

  Answer: Sorry for the confusion. Evaluation of disease degree is the direction in my study, yet we only focused on detection in this paper. Therefore, we changed the expression in Line110 in new version. Thank you for the correction.

Line107:  Please change ‘the current study’ to ‘this study’.

  Answer: Corrected. Thanks.

Line109: What do you mean about “a large-scale area”. Canopy scale? Farm-land scale? Continental scale?

 Answer: “a large-scale area” means that we can obtain the growth status of a large area orchard through UAV remote sensing.

Line117~120:  Please show some picture of sample D1 and D2 leaves. And how many trees were infected? Please explain the infected concentration and method. Further, please represent the number of total, infected, D1, and D2 trees as some Table.

 Answer: About the level of disease, we did get two levels of HLB. D1 and D2 were labeled by PCR test and naked eye observation. However, it is quite hard to discriminate the level in low-altitude by UAV remote sensing. Therefore, in this study, we only focus the discrimination between healthy category and HLB-infected one. We can neglect the difference on yellowness. Further, the number of infected tress was described in line 123-124.

Figure 1: How many times measured? Please represent your measurement schedule as some Table. And, this is very important one. Please show the time and weather condition of UAV measurement. According to measurement time, particularly in solar radiation and air temperature conditions, the performance of canopy spectral reflectance will be changed.

 Answer: Thank you for the comment. We did many times of measurement, almost once a month. However, we didn’t do many times in one day. In the UAV remote sensing experiment, it is not so easy to collect the data following the measurement schedule. The weather, the operation and the battery are all the problems, I think your suggestion is a very good one, we will make a measurement schedule in later experiment and try to conduct it.

Line127-136: Did you use Pix4D? or another SW? And, did you use only one grey board? How about black, white, and another grey board?

Answer: Pix4D is more suitable for stitching visible light and multi-spectrum. For the stitching scheme of this camera, Photoscan has more obvious advantages.

Before collecting data, S185 needs to calibrate with whiteboard and blackboard. This is the normal operation of the camera. This method is also introduced in other papers. This article does not explain this aspect too much.

Figure 2: ‘System’? What was the system?

Answer: ‘System’ is mean hyperspectral data acquisition device.

Figures 2 and 4: According to your picture, HH2 might measure the leaves at lower canopy. However, S185 measured the leaves at upper canopy. It may be one of the reasons on the different performance of S185 and HH2 in Figure 4(first_b).

Answer: Thank you for the explanation, we will add it in the paper.

Figure 4(second_b): What is ‘wirtebroad’?

Answer: Sorry for the clerical error, it should be whiteboard.

Figure 4(d,e): Which band was used for reflectance?

Answer: full band.

Figure 7: Please explain why smoothing was used. After smoothing, the spectral plot in the graph will be beautiful, but it will make more error.

 Answer: The purpose of using smoothing and denoising is to obtain a spectrum closer to the real spectrum, which does not cause greater errors.

Line248: Please write the library or module of python.

 Answer: Information about the algorithm of Isolated Forests can be found at the following website.https://github.com/scikit-learn/scikit-learn/blob/master/sklearn/ensemble/iforest.py

Line252: Sample number is enough for training? Please explain that. And did you use only S185 data? How about HH2?

Answer: The agriculture data especially UAV hyperspectral data is quite hard to capture a lot as the weather and operation factors. We did data acquisition almost monthly, however, the useful data is still not abundant. In this study, S185 data is captured by UAV and was used for HLB detection in a large scale, HH2 data only to used for evaluation of the S185 data.

3.2 and 3.4 : I don’t understand why you write these sections in this paper. It is not review paper. I cannot find any relations between your results and these two sections. Please explain that.

Answer: Section 3.2 and 3.4 describes the features extraction methods which are related closely to the results of classifiers. Section 3.2 introduces canopy spectral characteristic parameters in full bands, and section 3.4 describe the vegetation indices calculation method from extracted characteristic bands. Different features input to classifiers get different performance.

Round 2

Reviewer 2 Report

Dear authors, all my recommendations were adopted. There were some improvements on section 2 and the discussion is now more literature based than the previous version. The paper is fine for publication in my opinion.

Best regards,

Reviewer 4 Report

Even though you just answered to comments, the manuscript was not revised as following comments and your answers.